# Nutritional status, dietary habits, and their relation to cognitive functions: A cross-sectional study among the school aged (8–14 years) children of Bangladesh

Mowshomi Mannan Liza[1], Simanta Roy[1]*, Mohammad Azmain Iktidar[1], Sreshtha Chowdhury[1], Azaz Bin Sharif[1,2]

1 Department of Public Health, North South University, Dhaka, Bangladesh, 2 Global Health Institute, North South University, Dhaka, Bangladesh

☯ These authors contributed equally to this work.
* simantaroy23@gmail.com

## Abstract

### Background

Limited research addressed links between nutritional status, dietary habits, and cognitive functions in young children. This study assessed the status of cognitive functions and their association with nutritional status and dietary habits of school age children of Bangladesh.

### Methods

This cross-sectional multi-centre study was conducted on 776 participants in 11 conveniently selected educational institutions. A printed questionnaire with three sections (Section 1: background information, section 2: PedsQL™ Cognitive Functioning Scale, and section 3: semi-quantitative food-frequency questionnaire) was utilized for the data collection purpose. Sections 1 and 3 were self-reported by parents, and trained volunteers completed section 2 in-person along with the anthropometric measurements. Statistical analyses were done in Stata (v.16). Mean with standard deviation and frequencies with percentages were used to summarize quantitative and qualitative variables, respectively. Pearson's chi-square test and Spearman's rank correlation coefficient were used to explore bivariate relationships.

### Results

The mean age of the participants was 12.02±1.88 years, and the majority (67%) were females. The prevalence of poor cognitive function was 46.52%, and among them, 66.02% were females. In terms of body mass index (BMI), 22.44% possessed normal weight, 17.51% were overweight, and 5.19% were obese. This study found a statistically significant relationship between BMI and cognitive functions. Furthermore, different dietary components (e.g., protein, carbohydrate, fat, fiber, iron, magnesium) showed a significant (p<0.05 for all) weak positive correlation with cognitive function.

**Data Availability Statement:** All relevant data are within the manuscript and its Supporting Information files.

**Funding:** The author(s) received no specific funding for this work.

**Competing interests:** The authors have declared that no competing interests exist.

## Conclusion

BMI and dietary habits were associated with the cognitive function of young children in Bangladesh. Although the cross-sectional design of the study precludes causal relationships from being determined, the study finding deserves further examination via longitudinal research.

## Introduction

Child malnutrition is a critical global issue, accounting for approximately 11% of the global disease burden and over half of child deaths in developing countries [1]. More than 650 million people worldwide are overweight, and about 340 million children aged 5–19 years are overweight or obese [2]. The prevalence of childhood obesity has increased dramatically in metropolitan areas of Bangladesh, which was observed at less than 1% to 17.9% [3]. Also, micronutrient deficiencies are highly prevalent in children under five years, and it did not significantly improve over the past decade (2011–2021) [4].

The maturation and development of children's central nervous systems and cognitive function are substantially influenced by nutrition [5]. Also, exposure to educational institution and social interactions play a crucial role in this regard. Therefore, nutritional stability is vital throughout the school years, which is a time of intense growth due to increased physical and mental demands [6,7]. Proximate principles and micronutrients are more important for children because of their increasing dietary needs [8]. According to WHO, the recommended daily calorie intake is 2100 kcal per person, which varies based on age and sex [9]. Fat, protein, vitamin A, thiamine, riboflavin, niacin, folic acid, vitamin B12, vitamin C, and iodine are essential for children's growth and development, specifically, the calcium intake for children between the age of 10 to 14 is significant due to their rapid growth [9,10]. Micronutrient deficiencies, including iron, vitamin B12, folate, zinc, and vitamin D, are significantly associated with poor cognitive functions [11]. However, an estimated 25–27% of adolescents in Bangladesh are anaemic (Hb 12 g/dL), and 30% of 14–18-year-olds are iron-deficient (15%); as many as half of all school aged children (47–54%) do not have sufficient vitamin A [12]. Moreover, 60% of school children in Dhaka city aged 10 to 16 are not getting the recommended amount of protein, iron, and calcium [12]. Therefore, it is crucial to understand the effect of such deficiencies on the intellectual health of the children specifically on their cognitive functions.

Several prior studies tried to explain the effects of nutrition on the cognitive function of children. For example, a review article reported a correlation between obesity, cognitive and motor function declines, and altered brain plasticity [13]. In addition, nutritional deficiency can cause more absenteeism, earlier drop out, and poor performance in the classroom among the school aged children [14]. However, a prospective cohort study failed to determine any relationship between cognitive domain scores and nutritional status, although the primary focus of the study was patients with Alzheimer's disease and mild cognitive impairment [15,16]. Another study conducted in South India, indicates that cognitive function throughout middle childhood are vital in association with early-life malnutrition and future health. They evaluated cognitive achievements in reading and mathematics by questions with varying difficulty level to better understand their impact on cognitive functions [17]. However, to date there is no study conducted in Bangladesh to explore the relationship between nutrition and cognitive function, specifically among school aged children. Therefore, [17,18] the aim of this study was to investigate the relationship between the nutritional status, dietary habits, and cognitive functions among school aged children in Bangladesh.

## Methods

### Study design, setting, and sample

Children between the ages of 8 to 14, who were enrolled in grades 4 to 7 at five private schools, five public schools, and one Madrasah (a facility specifically designed for Islamic education and culture) in Bangladesh, were included in this cross-sectional study. Convenient sampling was used to select the institutions from three metropolitan cities of Bangladesh (Dhaka, Chattogram, and Cumilla) (Fig 1). All students from these institutions were considered for participation upon fulfilling the inclusion criteria. The study excluded participants with pre-existing cognitive decline, mental health conditions, or any other illness that can affect cognitive functions. Eligible participants underwent a face-to-face interview by trained volunteers, while the parent responses were collected using a self-reported questionnaire.

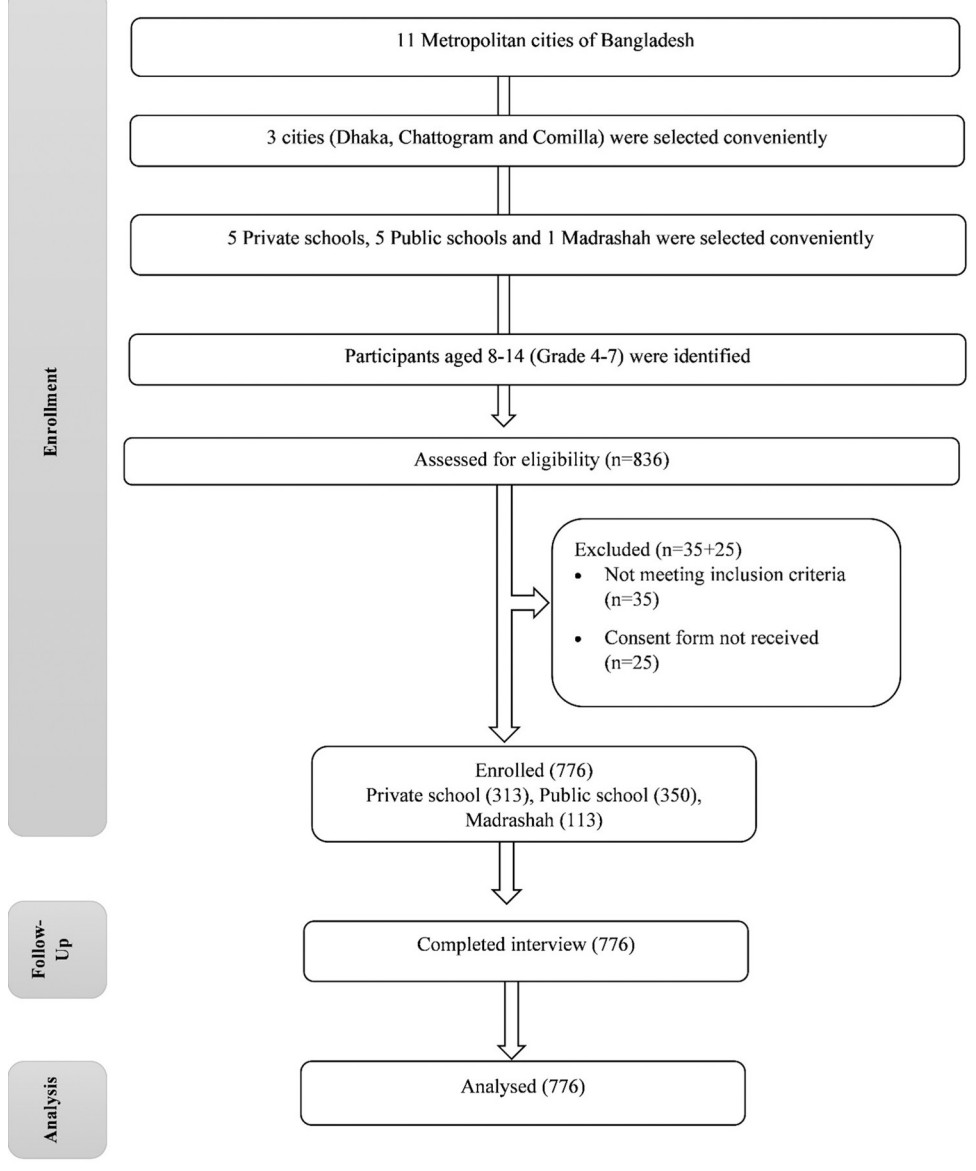

**Fig 1. CONSORT flow diagram.**

Informational brochures, parental permission forms, questionnaires, and instructions were distributed to every participant in the selected schools. The brochures included contact details of a study volunteer in case any question arises during the questionnaire completion process. Out of 836 eligible students, 776 children (response rate 92.8%) who submitted written parental consent and questionnaire within a week were considered for cognitive function assessment interviews and anthropometric measurements. Data were acquired from May to November of 2021 with approval from the Institutional Review Board of North South University (Approval no-2022/0R-NSU/IRB/1005). The study was carried out in compliance with the Helsinki Declaration 1964.

## Pretesting

Twenty participants from government and private schools each participated in pre-test to determine the feasibility and validity of this research. To accelerate data gathering without sacrificing data quality, necessary changes were made. On the recommendations of the pilot participants, the inclusion of the helpline number in brochures was taken into consideration.

## Measures

This study employed a semi-structured questionnaire with three parts. Part 1 included questions about age, gender, residence, family type, family income, parental education status, birth order and delivery method during birth, EPI vaccination status, duration of breastfeeding, and deworming status. Sections 2 and 3 contained the PedsQL[TM] Cognitive Functioning Scale [19,20], and a 39-item semi-quantitative food-frequency questionnaire (FFQ) [21], respectively. Both of these questionnaires were pre-validated and were used to evaluate cognitive functions and dietary intakes, respectively. Sections 1 and 3 were sent to the parents with clear instructions on how to complete them. A face-to-face interview of participants was conducted after they had returned with a completed questionnaire with parental consent. Trained volunteers completed the remaining portions of the questionnaire (section 2: PedsQL[TM] Cognitive Functioning Scale) with information directly obtained from the child via face-to-face interview. Additionally, anthropometric measurements were also taken during the interview.

## PedsQL[TM] cognitive functioning scale

The PedsQL[TM] Cognitive Functioning Scale consists of six questions. This scale was developed through focus group discussions, cognitive interviews, pre-testing, and field-testing measurement development techniques which was used in one of our prior studies [22,23]. A five-point likert scale was used to assess this scale, with 0 denoting never, 1 denoting nearly never, 2 denoting sometimes, 3 denoting often, and 4 denoting almost always. All responses were reverse-scored and then linearly translated to a 0–100 scale (0 = 100, 1 = 75, 2 = 50, 3 = 25, 4 = 0), in accordance with established scoring protocols. Any score below the mean was considered as poor cognitive functioning and higher scores indicated higher functioning.

## Nutritional status

During the data collecting process, trained data collectors took standardized anthropometric measures such as height, and weight. A commercial, non-elastic measuring tape was used to take height measurements. The participants were asked to stand with their heads held high, legs dangling loosely, and barefooted. The highest point of the head was indicated on large white cardboard affixed to the wall. The length from the floor to that point was calculated using tape. The height was rounded to the nearest 0.1 cm. Body weight was measured using an

electronic weight scale in whole numbers with and without shoes, coats, or other bulky accouterments. The measures were then transformed using WHO AnthroPlus software for assessing growth data for the age category of 5–19 years into BMI-for-age-Z-score (BAZ). Hence, BMI was calculated and categorized accordingly.

### Dietary status

A pre-validated questionnaire including the common food items was used to record the participant's dietary intake using a 7-day recall food diary [24]. Guardians were asked about the number of months per year they consumed each item, the number of days per week they ate the thing during those months, the number of times in a typical day they ate the item, and the amount they consumed on average each time. Pictures of different locally used plates and utensils were used to define the portion sizes. The nutritional value of the food items was calculated using the food composition table for Bangladesh [25].

### Statistical analysis

All analyses were performed using STATA MP Version 16. Descriptive statistics, such as mean, standard deviations, frequencies, and percentages were calculated. Association between two categorical variables was performed using the chi-squared test. The correlation between cognitive function and dietary nutrient intake was evaluated using Spearman's correlation coefficient. A p-value of <0.05 was considered statistically significant.

### Result

Background information on the study participants is presented in Table 1. The mean (±SD) age of the school-aged children between 8 and 14 years was 12.02 (±1.80) years. Among the 776 participants, the majority were female (67.31%), and lives in urban areas (61.48%). About 78% of the participants belong to a nuclear family, and most of them have birth orders first or second. Most of the participants' family income was 27,546.21 BDT or more. Around half of the responding guardians had less than 10 years of schooling experience. Almost 68% of the children were born to mothers via normal vaginal delivery (NVD). The rate of EPI vaccination was high (90.56%) among the respondents, and most of them were breastfed (73.28%) for more than a year. Majority of the participants either regularly (48.1%) or occasionally (49.08%) dewormed.

Table 2 demonstrates the nutritional status of the study participants. In the BMI category, 22.44% were underweight, 17.51% were overweight, and 5.19% were obese.

Average daily intakes of nutrients with their reference values from 7-day recall food diary are presented in Table 3. The table showed different nutrient deficiencies (e.g., fibre, iron, calcicum) to varied extend.

Table 4 shows that 46.5% of participants had a poor cognitive function, and the remaining 53.48% had a good cognitive function. Among them, females with poor cognitive function, 66.02%, and their mean ages were 12.02±1.88 and 12.0±1.66. Most of the participants were from urban areas with poor and good cognitive function 61.5% and 55.9%, respectively, and in the case of nuclear families' participants with poor cognitive function were 78.65%. Roughly 74% had first and second birth orders with poor cognitive function. Around half of the responding guardians had poor cognitive functional children. For most of the participants, around 70% had poor cognitive function; the mode of delivery was NVD. Around 91% of the participants were EPI-vaccinated with poor cognitive function. Almost half (47.59%) of the participants with poor cognitive function were breastfed for more than 24 months. Participants with poor cognitive function had 52.68% dewormed occasionally. Children with poor

**Table 1. Background information of study participants (n = 776).**

| Variables | Frequency | Percentage |
|---|---|---|
| **Age (in years), mean±SD** | 12.02±1.80 | |
| **Gender** | | |
| Male | 254 | 32.69 |
| Female | 522 | 67.31 |
| **Residence** | | |
| Rural | 299 | 38.52 |
| Urban | 477 | 61.48 |
| **Type of family** | | |
| Nuclear | 602 | 77.59 |
| Joint | 174 | 22.41 |
| **Birth Order** | | |
| 1st or second | 574 | 74.01 |
| Third or more than third | 202 | 25.99 |
| **Monthly family income (in BDT) mean±SD** | 27546.21± 66198.76 | |
| **Guardians' education** | | |
| Less or Equal to 10 years | 407 | 52.5 |
| More than 10 years | 369 | 47.5 |
| **Mode of delivery** | | |
| Do not know | 45 | 5.86 |
| NVD by others | 185 | 23.8 |
| NVD by Doctor | 342 | 44.02 |
| Caesarean section | 204 | 26.32 |
| **EPI vaccination** | | |
| No | 73 | 9.44 |
| Yes | 703 | 90.56 |
| **Duration of breastfeeding (in months)** | | |
| < 6 months | 84 | 10.78 |
| 6–12 months | 124 | 15.93 |
| 12–24 months | 198 | 25.49 |
| >24 months | 371 | 47.79 |
| **Deworming** | | |
| Never | 22 | 2.82 |
| Occasionally | 381 | 49.08 |
| Regularly (3 monthly) | 373 | 48.1 |

SD, standard deviation; BDT, Bangladeshi taka, NVD, Normal Vaginal Delivery.

**Table 2. Nutritional status of study participants.**

| Variables | Frequency | Percentage |
|---|---|---|
| **BMI (Body Mass Index) (n = 771)** | | |
| Underweight (<-2SD) | 173 | 22.44 |
| Normal weight (< +1SD to < -2SD) | 423 | 54.86 |
| Overweight (>+1SD) | 135 | 17.51 |
| Obese (>+2SD) | 40 | 5.19 |

**Table 3. Average daily intake of dietary nutrients with their reference value 7-day recall food diary.**

| Nutrients | Reference Value | Mean±SD | Nutrients | Reference Value | Mean±SD |
|---|---|---|---|---|---|
| Energy (Kcal) | 35 to 65 kcal/kg[26] | 913.38±310.24 | Vitamin A (µg) | 600 [27] | 606.43±391.35 |
| Protein (g) | 0.8g/kg[28] | 47.47±25.99 | Vitamin E (mg) | 11 [27] | 2.65±1.71 |
| Carbohydrate (g) | 191.7 ± 58.9[29] | 81.75±52.75 | Vitamin D (µg) | 15 [27] | 1.33±0.86 |
| Fat (g) | - | 12.56±7.30 | Vitamin C (mg) | 45 [27] | 40.76±26.30 |
| Dietary Fibre (g) | 25-31g[30] | 9.45±6.10 | Thiamine (mg) | 0.9 [27] | 0.50±0.31 |
| Iron (mg) | 8[27] | 6.95±4.18 | Riboflavin (mg) | 0.9 [27] | 0.72±0.45 |
| Calcium (mg) | 1300 [27] | 509.66±267.94 | Niacin (mg) | 12 [27] | 15.91±8.63 |
| Magnesium (mg) | 240 [27] | 139.04±85.94 | Pyridoxine (mg) | 1 [27] | 0.77±0.46 |
| Phosphorus (mg) | 1250 [27] | 614.74±346.02 | Folate (µg) | 600 [27] | 110.15±71.09 |
| Sodium (mg) | 1200 [27] | 352.18±198.05 | | | |
| Potassium (mg) | 2500 [27] | 1139.58±650.72 | | | |
| Zinc (mg) | 8 [27] | 7.39±4.42 | | | |
| Copper (mg) | 0.7 [27] | 1.13±0.63 | | | |

cognitive function had a body mass index (BMI) distribution as 23.5% underweight, 21.08% overweight, and the reamining 3.31% obese.

There was a significant but weak positive correlation between Cognitive function and the intake of protein, carbohydrate, fat, B vitamins, some water-soluble (Thiamine, Riboflavin, Niacin, Pyridoxine, Folate), fat-soluble vitamins (Vitamin A, D, E), and minerals including iron, magnesium, Phosphorus, Sodium, Potassium, zinc, Copper ($p < 0.05$) (**Table 5**).

## Discussion

The study's results emphasize the significant correlation between BMI and cognitive function in school aged children. Although it is difficult to isolate the specific impact of nutrition due to its intricate interaction with factors such as demography, socioeconomic position, health, and genetic influences, this study supports previous research that emphasizes the crucial role of nutrition in cognitive development [16,31,32]. This study also corroborates global patterns of childhood obesity and overweight, consistent with findings from Dhaka city, which indicate the increased prevalence of overweight and obesity among school-going children [33]. The results of this study indicate that 46.52% of the participants showed poor cognitive function, which is consistent with a study conducted in Bangladesh. The finding highlights the importance of raising parental awareness regarding the cognitive health of children in Bangladesh.

The results of this study found that among school aged children, in terms of BMI, 3.31% obese and 21.08% overweight children had poor cognitive functions. Similarly, a study on children aged between 9–10 years old also reported a potential link between BMI and cognitive function [34]. Another study on young medical students emphasized a correlation between BMI and cognitive function [35]. This study also provides support that BMI is associated with cognitive functions. The possible mechanism can include changes in cortical volumes and other morphological alterations, especially in regions of the prefrontal cortex responsible for executive function [36]. However, a better understanding of this issue can be achieved through a nationwide study on a diverse population with the help of clinical examination of cognitive function.

Several prior reviews highlighted the role of different nutrients in the development and proper functioning of brain [10,36]. For a better cognitive function, it is necessary to consume right amounts of protein and energy, in addition to the essential micronutrients [37]. Using a sizable sample, this study found deficiencies in a number of nutrients and different dietary nutrients (e.g., protein, carbohydrate, fat, iron, calcium) also showed a very weak positive

**Table 4. Cognitive function of the study participants (n = 776).**

| Variables | Poor Cognitive Function (n = 361, 46.52%) | Good Cognitive Function (n = 415, 53.48%) | p-value |
|---|---|---|---|
| | Frequency (%) | Frequency (%) | |
| **Age (in years), mean±SD** | 12.02±1.88 | 12.05±1.66 | 0.61[Ψ] |
| **Gender** | | | |
| Male | 123 (33.98) | 118(28.47) | 0.09[€] |
| Female | 238 (66.02) | 297(71.53) | |
| **Residence** | | | |
| Rural | 139 (38.5) | 183(44.1) | 0.12[€] |
| Urban | 222 (61.5) | 232(55.9) | |
| **Type of family** | | | |
| Nuclear | 284 (78.65) | 321(77.34) | 0.66[€] |
| Joint | 77 (21.35) | 94(22.66) | |
| **Birth Order** | | | |
| 1st or second | 265 (73.82) | 293 (73.62) | |
| Third or more than third | 94 (26.18) | 105 (26.38) | 0.95[€] |
| **Monthly family income (in BDT) mean±SD** | 27546.21± 66198.76 | | 0.50 [Ψ] |
| **Guardians' education** | | | |
| Less or Equal to 10 years | 187 (51.9) | 234(56.34) | 0.31[€] |
| More than 10 years | 174 (48.1) | 181(43.66) | |
| **Mode of delivery** | | | |
| Do not know | 16 (4.43) | 26 (6.27) | 0.58[€] |
| NVD by others | 94 (26.04) | 99(23.86) | |
| NVD by Doctor | 157 (43.49) | 189(45.54) | |
| C/S | 94 (26.04) | 101(24.34) | |
| **EPI vaccination** | | | |
| No | 31 (8.59) | 43 (10.36) | 0.4[€] |
| Yes | 330 (91.41) | 372(89.64) | |
| **Duration of breastfeeding** | | | |
| < 6 months | 43 (11.9) | 41 (9.98) | 0.77[€] |
| 6–12 months | 55 (15.3) | 72 (17.27) | |
| 12–24 months | 91 (25.21) | 102 (24.57) | |
| >24 months | 172 (47.59) | 200 (48.18) | |
| **Deworming** | | | |
| Never | 9 (2.54) | 14 (3.46) | 0.13[€] |
| Occasionally | 190 (52.68) | 189(45.43) | |
| Regularly (3 monthly) | 162 (44.79) | 212(51.11) | |
| **BMI (body mass index) Category** | | | |
| Underweight | 78 (23.5) | 83 (51.55) | |
| Normal weight | 173 (52.11) | 228(58.31) | *0.048*[€] |
| Overweight | 70 (21.08) | 55(14.07) | |
| Obese | 11 (3.31) | 25(6.39) | |

SD, standard deviation; BDT, Bangladeshi taka; NVD, Normal Vaginal Delivery

Ψ, Spearman correlation p-value

€, Chi-square test p-value.

**Table 5. Spearman's correlation between Cognitive function and daily nutrients intake.**

| Nutrients | r* | p-value** | Nutrients | r* | p-value** |
|---|---|---|---|---|---|
| Energy (Kcal) | 0.006 | **0.86** | Vitamin A (μg) | 0.1 | **0.02** |
| Protein (g) | 0.1 | **0.02** | Vitamin E (mg) | 0.1 | **0.02** |
| Carbohydrate (g) | 0.1 | **0.02** | Vitamin D (μg) | 0.1 | **0.02** |
| Fat (g) | 0.1 | **0.03** | Vitamin C (mg) | 0.1 | **0.02** |
| Dietary Fibre (g) | 0.1 | **0.02** | Thiamine (mg) | 0.1 | **0.03** |
| Iron (mg) | 0.1 | **0.03** | Riboflavin (mg) | 0.1 | **0.03** |
| Calcium (mg) | 0.03 | **0.37** | Niacin (mg) | 0.1 | **0.02** |
| Magnesium (mg) | 0.1 | **0.03** | Pyridoxine (mg) | 0.1 | **0.03** |
| Phosphorus (mg) | 0.1 | **0.03** | Folate (μg) | 0.1 | **0.02** |
| Sodium (mg) | 0.1 | **0.03** | | | |
| Potassium (mg) | 0.1 | **0.03** | | | |
| Zinc (mg) | 0.1 | **0.03** | | | |
| Copper (mg) | 0.1 | **0.02** | | | |

* r is Spearman's correlation coefficient.

** p-value is for Spearman's correlation test, p < 0.05 is statistically significant.

significant correlation with cognitive function. The self-reported data on dietary intake and non-clinical methods of cognitive function assessment can be a reason for this weak relationship. Still, this finding can be a representation of the bigger picture which can be assessed in a larger study [37,38]. Therefore, further study is required to investigate the consequences of dietary consumption on cognitive function more precisely.

To our best knowledge, this is the first study ever conducted in Bangladesh to investigate the association between cognitive function, nutritional status, and dietary intake among school-going children. The questionnaires used in this research are more reliable since they were adapted from previously validated scales. The large sample size also helps ensure the reliability and validity of the results. However, this study has some limitations. First, the cross-sectional design and unavailability of multivariate analysis preclude causal relationships from being determined. However, it deserves further exploration through longitudinal studies. Larger studies and clinical trials examining the effect of different nutrients on the cognitive function of young children are warranted. Second, the measurement of cognitive function may not be accurate, considering the absence of clinical tests. Third, there is a higher risk of recall biases in terms of FFQ. Also, the convenience sampling techniques may cause selection bias and limit the generalizability of the study results.

## Conclusion and recommendation

According to the results of this research, this study found an association between BMI and cognitive function and identified several nutrients weakly associated with the cognitive function of school aged children. However, we could not draw any direct association between cognitive function, dietary habits, and nutritional status. Further longitudinal studies are required to draw a better conclusion.

## Supporting information

**S1 Checklist. STROBE statement—Checklist of items that should be included in reports of observational studies.**
(DOCX)

**S1 Dataset.**
(XLSX)

## Acknowledgments

The authors would like to convey special thanks to the reviewers for their inputs to improve the quality of this manuscript. The authors would also like to thank Humayara hema (Chattagram Medical College and Hospital), Tabassum Mustary (Chattagram Medical College and Hospital), Tarannum Mahmud (Chattagram Medical College and Hospital) for their assistance and time during the development of this manuscript.

## Author Contributions

**Conceptualization:** Mowshomi Mannan Liza, Simanta Roy, Mohammad Azmain Iktidar, Sreshtha Chowdhury, Azaz Bin Sharif.

**Data curation:** Mowshomi Mannan Liza, Sreshtha Chowdhury.

**Formal analysis:** Simanta Roy, Mohammad Azmain Iktidar.

**Investigation:** Mowshomi Mannan Liza, Mohammad Azmain Iktidar.

**Methodology:** Simanta Roy.

**Project administration:** Mowshomi Mannan Liza, Simanta Roy, Mohammad Azmain Iktidar.

**Resources:** Mowshomi Mannan Liza, Simanta Roy, Mohammad Azmain Iktidar, Sreshtha Chowdhury.

**Software:** Mohammad Azmain Iktidar, Azaz Bin Sharif.

**Supervision:** Mowshomi Mannan Liza, Simanta Roy.

**Visualization:** Simanta Roy.

**Writing – original draft:** Mowshomi Mannan Liza, Simanta Roy, Sreshtha Chowdhury.

**Writing – review & editing:** Mowshomi Mannan Liza, Mohammad Azmain Iktidar, Azaz Bin Sharif.

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
