## [Decision Letter · Decision Letter 0]

14 Dec 2023

PONE-D-23-15685Nutritional Status, Dietary habits, and their Relation to Cognitive Function: A Cross-sectional Study among the School-Going Children of BangladeshPLOS ONE

Dear Dr. Roy,

Thank you for submitting your manuscript to PLOS ONE. After careful consideration, we feel that it has merit but does not fully meet PLOS ONE’s publication criteria as it currently stands. Therefore, we invite you to submit a revised version of the manuscript that addresses the points raised during the review process.

We look forward to receiving your revised manuscript.

Kind regards,

Sanjoy Kumer Dey, M.D

Academic Editor

PLOS ONE

Journal Requirements:

4. We notice that your supplementary figure 1 is uploaded with the file type 'Figure'. Please amend the file type to 'Supporting Information'. Please ensure that each Supporting Information file has a legend listed in the manuscript after the references list.

Reviewers' comments:

Reviewer's Responses to Questions

**Comments to the Author**

1. Is the manuscript technically sound, and do the data support the conclusions?

Reviewer #1: Partly

Reviewer #2: Partly

2. Has the statistical analysis been performed appropriately and rigorously? 

Reviewer #1: No

Reviewer #2: Yes

3. Have the authors made all data underlying the findings in their manuscript fully available?

Reviewer #1: Yes

Reviewer #2: Yes

4. Is the manuscript presented in an intelligible fashion and written in standard English?

Reviewer #1: No

Reviewer #2: Yes

5. Review Comments to the Author

Reviewer #1: This cross-sectional study focused on dietary habits, nutritional status and cognition among children in Bangladesh. Relations/relationships are generally identified through regression. As there were 776 children included within the study unsure why only correlations were performed. There were areas that needed to be rewritten for grammar and syntax. The cognitive functioning scale appears more for ADHD or other behavioral disorders as opposed to cognitive status based on taking a test as is done with other assessments. As no other measurements were assessed (e.g. status in school or test scores), it is challenging to state cognitive status was poor or good. Even though stunting and weight are considered ways to measure nutrition, they are not the only ones and are more for anthropometrics. Curious if skinfolds or other techniques were used to confirm nutritional status.

Title: School-going may need to be revised to possibly School Children or Elementary or specific age groups of the children who were included in this study.

Abstract: in the methods mentioned qualitative information, none were included within the results. Curious if qualitative variables were used in a different context than actual interview or text-based data obtained through the interviews, please clarify. For the results, indicate if the association between BMI and cognitive function were positive or negative. Clarify which or provide examples of those nutrients that were associated with cognitive function.

Introduction:

In the first paragraph, expand if it is all during elementary years or only during early childhood, etc. The first three paragraphs of the introduction can be condensed and synthesized for clarity as essentially the information from these paragraphs are the same.

Cognitive functions are different dependent on the growth of the child, so young children <5 years will have different functions that a child who is 6-12 years old and even different between that age group. Please expand on the age that is the focus and what other studies in developing countries have shown this relationship between cognition and nutrition and why this study is important for those in Bangladesh. Please be sure to write the research question as open ended as opposed to yes/no

Methods:

For the pre-testing, expand on who these participants consisted of – children or other age groups. Also, indicate how 20 was appropriate for this pre-testing (e.g. reference).

Line 135: it should be bare foot instead of ‘naked feet’

Line 137: indicate if body weight was taken to the tenth or whole number.

For the statistical analysis indicate the statistical significance.

For the public involvement, unsure the purpose of this portion as mentioned in the pretesting how this was achieved. How were parents involved in the study design and why? Please clarify this portion.

Results:

For table 2, thin is not the classification based on BMI, it would be underweight.

For table 4, please place the actual p-values and not just <0.05.

For dietary intake, it would be relevant to include the actual nutrients to determine if these children were consuming less than the requirements as opposed to just including the correlation values.

Discussion:

In the first sentence, mentions that BMI was statistically significant with cognitive function, yet the p-value was 0.05, so it appears there was no significance as p<0.05 appeared to be the mechanism to detect significance.

Much of the discussion included results, which can be eliminated and instead focus on why these results were similar or not to other studies.

For references that were included, may indicate how cognition was measured as each study uses a different measurement and it would be important to indicate how they differ.

As the outcomes were focused on dietary, there was sparse information in the discussion that focused on these results and why.

Reviewer #2: It is an existing knowledge that poor nutrition and dietary intake interfere with cognitive development in children. Though the study is the first ever study (according to author declaration), it was better to draw some novel inference from the findings and discuss accordingly.

6. PLOS authors have the option to publish the peer review history of their article (what does this mean?). If published, this will include your full peer review and any attached files.

Reviewer #1: No

Reviewer #2: **Yes: **Md Kamruzzaman

---

## [Author Response · Author response to Decision Letter 0]

30 Jan 2024

Thank you for the review. The manuscript has been updated with a thorough grammar check. We have attached a point-by-point response to all the comments.

---

## [Decision Letter · Decision Letter 1]

13 May 2024

Nutritional Status, Dietary Habits, and their Relation to Cognitive Functions: A Cross-sectional Study among the School Aged (8-14 years) Children of Bangladesh

PONE-D-23-15685R1

Dear Dr. Simanta Roy,

We’re pleased to inform you that your manuscript has been judged scientifically suitable for publication and will be formally accepted for publication once it meets all outstanding technical requirements.

Kind regards,

Sanjoy Kumer Dey, M.D

Academic Editor

PLOS ONE

Additional Editor Comments (optional):

Reviewers' comments:

Reviewer's Responses to Questions

**Comments to the Author**

1. If the authors have adequately addressed your comments raised in a previous round of review and you feel that this manuscript is now acceptable for publication, you may indicate that here to bypass the “Comments to the Author” section, enter your conflict of interest statement in the “Confidential to Editor” section, and submit your "Accept" recommendation.

Reviewer #1: All comments have been addressed

Reviewer #2: All comments have been addressed

2. Is the manuscript technically sound, and do the data support the conclusions?

Reviewer #1: Yes

Reviewer #2: Yes

3. Has the statistical analysis been performed appropriately and rigorously? 

Reviewer #1: Yes

Reviewer #2: Yes

4. Have the authors made all data underlying the findings in their manuscript fully available?

Reviewer #1: Yes

Reviewer #2: Yes

5. Is the manuscript presented in an intelligible fashion and written in standard English?

Reviewer #1: Yes

Reviewer #2: Yes

6. Review Comments to the Author

Reviewer #1: The authors addressed all aspects of the reviewer's comments and has significantly improved in readability. One line 242, remaining is misspelled, so please correct that information.

Reviewer #2: The author addressed all comments properly raised in the 1st review process. In my opinion the manuscript is now suitable to publish.

7. PLOS authors have the option to publish the peer review history of their article (what does this mean?). If published, this will include your full peer review and any attached files.

Reviewer #1: No

Reviewer #2: **Yes: **Md Kamruzzaman

---

## [Editor Report · Acceptance letter]

16 May 2024

PONE-D-23-15685R1 

PLOS ONE

Dear Dr. Roy, 

I'm pleased to inform you that your manuscript has been deemed suitable for publication in PLOS ONE. Congratulations! Your manuscript is now being handed over to our production team.

Kind regards, 

on behalf of

Dr. Sanjoy Kumer Dey 

Academic Editor

PLOS ONE